# Understanding the Factors Influencing Cat Containment: Identifying Opportunities for Behaviour Change

**DOI:** 10.3390/ani13101630

**Published:** 2023-05-12

**Authors:** Gemma C. Ma, Lynette J. McLeod

**Affiliations:** 1Sydney School of Veterinary Science, The University of Sydney, Camperdown, NSW 2006, Australia; 2Royal Society for the Prevention of Cruelty to Animals New South Wales, Yagoona, NSW 2199, Australia; 3School of Psychology, Speech & Hearing, The University of Canterbury, Christchurch 8041, New Zealand; lynette.mcleod@canterbury.ac.nz

**Keywords:** cat, containment, roaming, behaviour change, audience segmentation

## Abstract

**Simple Summary:**

Cats are popular pets in Australia, being present in around one-third of households. As pets, cats are managed in a wide variety of ways, from fully indoors in apartments to completely outdoor free roaming. Australian wildlife is uniquely vulnerable to cat predation. Roaming cats also create a nuisance and are at risk of accidents and injuries. Councils, veterinarians, animal welfare organisations and conservation groups all have an interest in encouraging cat owners to change their behaviour and prevent their cats from roaming. Understanding what influences cat owner decisions can help design effective programs. This study asked cat owners about their cats, living circumstances, current cat management behaviour and agreement with statements reflecting their ability to contain their cats and their social opportunity and motivation to do so. More than half of participating cat owners already fully contain their cats. The most important influence for cat owners to keep their cats contained was having the skills, knowledge and belief that they could do so successfully. Those who lived in apartments, were renting or were motivated by their cat’s safety, to protect wildlife or to care for their community were also more likely to contain their cats.

**Abstract:**

There are over 5 million pet cats in Australia managed on a spectrum from fully indoors to completely outdoor free roaming. Roaming cats threaten biodiversity, can create a nuisance and are at risk of accidents and injury. Hence, there is substantial interest in behaviour change interventions to increase cat containment. An online questionnaire collected information on cat owner demographics, the number of cats owned, current containment behaviours and an agreement with 15 capability, opportunity and motivation (COM) items. Responses were received from 4482 cat owners. More than half (65%) indicated that they currently keep their cat(s) fully contained. Another 24% practiced a night curfew. Owners’ psychological capability had the greatest influence on containment behaviour. Motivation (community- and cat welfare-framed), living in an apartment and renting were also associated with a greater likelihood of containment. Cat owners not currently containing their cats could be divided into six profiles who differed on agreement with COM themes, age, future intentions, current behaviour, location and gender. Understanding differences between cat owner segments can assist with designing behaviour change interventions. Increasing cat owners’ psychological capability to contain their cats and encouraging the adoption of a night curfew as a first step towards 24 h containment are recommended.

## 1. Introduction

Cats are valued companions and family members to many people all over the world. Almost one-third of Australian households are home to one or more cat, with an estimated population of more than 5.3 million pet cats nationally [1]. However, pet cats are managed by their owners in vastly different ways, from living fully indoors in apartments to completely outdoor free roaming and everything in between.

Australian wildlife is uniquely vulnerable to predation by cats, having evolved without exposure to felids. Cats were introduced to the island continent at the time of European colonisation and are thought to have since caused or contributed to the extinction of more than 30 mammalian species [2,3,4]. Cats mostly predate mammals but will also predate birds, reptiles and amphibians. The type and amount of prey will depend on prey abundance, and the cat’s individual preference [5,6,7]. Research suggests that most pet cats will hunt if given the opportunity, even if well fed, and that only a proportion of what is caught is brought home [5,8,9,10]. However, the composition of the food may play an important role in reducing hunting; increasing the meat content of food, along with providing opportunities to engage in predatory play, have been shown to reduce the predation of wild animals by pet cats [11]. Pet cats are estimated to predate substantially fewer animals compared with a wild free-living or ‘feral’ cat; however, the greater population density of pet cats has been estimated to result in wildlife impacts 18–50 times higher per square kilometre [8]. This represents an important threat to biodiversity, given that urban areas can provide valuable habitats for threatened species [12].

Free roaming also poses risks to cat safety. Trauma is a leading reason for pet cats to present to veterinary practices, especially due to road traffic accidents [13], and mortality rates from road traffic accidents can be 60% or higher [14]. Roaming cats are frequently injured when attacked by other cats or dogs [15] and are at risk of contracting infectious diseases such as feline immunodeficiency virus [16]. Roaming cats can also be a nuisance to neighbours through noise, property damage, soiling with urine and faeces and disturbing other companion animals.

A variety of stakeholders have an interest in encouraging a greater uptake of cat containment by cat owners, including wildlife and conservation organisations, local councils, animal welfare organisations, veterinarians and the non-cat owning public. A twenty-four-hour containment of cats to their owner’s property, either exclusively indoors or with controlled outdoor access, is recommended by RSPCA Australia to prevent the potential negative consequences of roaming [17]. In addition, mandatory containment or ‘cat curfews’ are increasingly being legislated across Australia; in a large 2020 survey, close to one-third of the 250 participating Australian local governments had either a cat curfew, 24 h containment or cat prohibition zones (or a combination of these) in place [18]. More recently, the Australian Capital Territory has required all cats born from 1 July 2022 onwards to be contained to owner premises 24 h a day [19].

A range of factors have been identified that influence cat containment behaviour, including owner demographics (age, gender, education and area of residence), owner ability to successfully contain their cat, their beliefs and attitudes around a cat’s rights to roam and their predation of wildlife, their concerns for cat safety, social norms around cat containment, as well as the cat’s characteristics (such as sex, breed, health and behaviour) [20,21,22,23,24,25,26,27,28,29]. Understanding these factors is the first step in designing more effective and targeted human behaviour change interventions.

Changing human behaviour can be difficult. Education and providing information are rarely enough [30]. Social psychology and behavioural economics have generated an array of models and frameworks designed to increase audience understanding, engagement and, ultimately, the adoption of desired behaviours. Two such frameworks that have been applied to domestic cat management include Community-based Social Marketing [31,32,33] and the Behaviour Change Wheel (BCW) [20,25,26,33,34]. This later framework, along with its associated integrative Capability-Opportunity-Motivation (COM) Behaviour model, was initially developed by Michie and her colleagues for application in the health field. It conceptualises influential factors into three categories:(1)Capability: An individual’s physical and psychological ability to perform a behaviour. For example, does the cat owner have the physical skills or knowledge and cognitive skills to contain their cat. Interventions that increase capability incorporate techniques that educate, train and provide personal support.(2)Opportunity: The physical and social factors external to an individual that prompt or enable a behaviour to occur. For example, does the cat owner have access to the relevant containment resources, and do they have support from family, neighbours and community to keep their cat contained. Interventions that increase opportunity incorporate techniques that: provide access, enable, facilitate, prompt or constrain.(3)Motivation: Factors internal to an individual that energise or direct behaviour. These factors can be either reflective (incorporating conscious deliberation and reasoning) or automatic (usually outside conscious control, e.g., impulse, habitual or emotional) [35]. For example, a cat owners’ decision to contain their cat may occur after careful cost-benefit deliberation, after their cat experienced a traumatic traffic incident or because that is what they have always done. Interventions that increase motivation incorporate techniques that: inform, persuade, discuss, demonstrate, incentivise or coerce.

Another important aspect in improving the behavioural impact of an intervention is matching its content to a specific audience need. Not every cat owner views their cats’ needs and their management approaches in the same way; thus, the patterns of the drivers of and barriers to cat containment will vary across individuals. Interventions can be designed or targeted to best match the characteristics of segments with specific driver/barrier profiles, and messages can also be crafted for specific individuals, as opposed to larger segments [36,37].

The objectives of this study were to: (1) document the containment behaviour of cat owners across New South Wales (NSW), (2) identify the main drivers and barriers to cat containment and organise them according to the COM Behavioural model, (3) determine the importance of these COM items, along with other demographic/situational variables in influencing current containment behaviour and future intentions of cat owners to restrict their cats roaming and (4) segment cat owners using these COM items and identify leverage points that may be useful for targeting interventions and informing policy and legislation.

## 2. Materials and Methods

### 2.1. Questionnaire

An online questionnaire was developed and advertised throughout NSW, with links available through the RSPCA NSW website and social media and shared by other external stakeholders, including veterinary clinics and councils throughout NSW (Appendix A). NSW residents aged 18 years and over were eligible to participate in the study. Only responses from current cat owners were included in the analysis. The human ethics approval was obtained from the University of Sydney’s Human Research Ethics Committee (Project Number 2021/473).

The questionnaire collected basic demographic information (age and gender) from all respondents. Cat owners were asked about the number of cats they owned, their current cat containment behaviours, their likelihood of future adoption of cat containment and an estimate of the time their cat currently spent roaming freely outside across specific parts of the day (which were combined to gain a total time spent outside freely roaming). They were asked about the characteristics of their home (location, type of dwelling, access to an outside space and home ownership) that had the potential to influence containment behaviour (physical opportunity). In addition, they were asked to rate their agreement (on a 5-point Likert scale) to 15 capability, social opportunity and motivation (COM) items relating to cat containment (Table 1). These COM items addressed four important themes that had been identified from a review of previous research: cat owners’ capability to contain their cat, social opportunity for cat containment, motivation for containment associated with improving cats’ welfare and motivation for containment associated with supporting the community [26,28,29,37].

### 2.2. Questionnaire

As COM items were worded as either drivers or barriers in the survey, all barrier items were reversed-scored for analysis. All data was tested for compliance with the assumptions for parametric statistical analyses: normality, outliers, multicollinearity, non-linearity, homoscedasticity and non-independence assumptions. COM agreement data from Likert scales were treated as interval data, following the common practice used in medical and psychological research [38]. Internal consistency of the COM variables containing multiple items was tested using the Cronbach’s Alpha Test [39].

ANOVAs and Pearson Chi-Squared were used to compare the differences in demographic and situational variables between various groups of respondents. A multiple regression was performed to identify the variables associated with cat free-roaming behaviour (measured as time spent outside freely roaming). A Latent Profile analysis (LPA) was conducted to classify cat owners into homogenous segments based on their responses to the COM items. The relative model fit was assessed using the Bayesian information criteria (BIC) [40] relative entropy [41] and the Lo–Mendell–Rubin likelihood ratio test (LMR) [42]. A significant *p* value from the LMR test (*p* < 0.05) indicated that the given profile solution fit the data significantly better than the solution with one fewer profile groups. MANOVAs, ANOVAs and Pearson Chi-Squared were used to compare the differences in demographic and situational variables between the cat owner LPA profiles. All analyses were conducted using SPSS version 26.0 (IBM, Armonk, NY, USA, 2019) except for the LPA which was conducted in MPlus version 8.9 (Muthén and Muthén, Los Angeles, CA, USA, 2019).

## 3. Results

### 3.1. Participant Characteristics

The online questionnaire received 4482 complete responses from cat owners from 105 of the 128 NSW local government areas. Over two-thirds of respondents (71%: 3177) lived in major urban centres (Sydney, Newcastle, Lake Macquarie, Central Coast, Wollongong and Shoalhaven), with the remaining third coming from regional areas [43].

The overall average age of respondents was 46.1 years (±13.7). Over three quarters of respondents were female (3684, 80%), with 601 males (13%) and 186 identifying as non-binary (4%). As a response to this question was not compulsory, 11 respondents chose not to offer a response.

Just over half of the respondents (2325; 52%) owned one cat, a third (1453; 33%) owned two cats, and 665 (15%) owned 3 cats or more (Table 2). More than half of the cat owners (2896 of 4482; 65%) indicated that they currently keep their cat(s) fully contained on their property (24 h contain), either indoors all the time (1606, 36%) or with restricted outdoor access using a cat enclosure or on a lead (1290, 29%). A further 1088 cat owners (24%) practiced a night curfew, whereby they kept their cat(s) indoors during the night but allowed their cats to roam freely during the day. The remaining 492 cat owners (11%) allowed their cats to roam freely during the day and night (24 h roam).

Comparisons between cat containment behaviour and demographic, situational and cat ownership variables demonstrate that:Cat owners who practiced a night curfew were significantly older (mean 48.9 years) than both owners who 24 h contained their cats (mean 45.2 years) and owners who allowed their cats to 24 h roam (mean 44.8 years) (F = 29.41, df = 2, *p* < 0.001, η^2^ = 0.01).Male cat owners were more likely to allow their cat(s) to 24 h roam than female owners (15% vs. 10%; Pearson Χ^2^ = 14.01, df = 4, *p* = 0.01, r = 0.04).There was no statistical difference in containment behaviours between urban and regional locations (Pearson Χ^2^ = 5.08, df = 2, *p* = 0.08, r = 0.03).Cat owners who rented were more likely to 24 h contain their cat than those who owned their home (74% vs. 62%; Pearson Χ^2^ = 46.42, df = 2, *p* < 0.001, r = 0.09).Owners living in an apartment or unit were more likely to 24 h contain their cat than owners living in free-standing houses (82% vs. 59%; Pearson Χ^2^ = 182.80, df = 2, *p* < 0.001, r = 0.18).Owners without access to outdoor spaces at their homes were more likely to 24 h contain their cat than owners with access (73% vs. 63%; Pearson Χ^2^ = 25.75, df = 2, *p* < 0.001, r = 0.07).Cat owners who owned five cats or more were more likely to 24 h contain their cats compared to cat owners who owned one cat (75% vs. 62% Pearson Χ^2^ = 23.19, df = 8, *p* = 0.003, r = 0.05).

Those cat owners who allowed their cats to roam away from their property (i.e., night curfew and 24 h roam) were asked to estimate the time their cat currently spent outside. These cat owners were also asked about their future intentions of 24 h containment. Cats who were allowed to 24 h roam spent significantly more time outdoors than those under a night curfew (Figure 1; Pearson Χ^2^ = 158.02, df = 3, *p* < 0.001, r = 0.27). Cat owners currently practicing a night curfew were more likely to have future intentions of preventing their cat from roaming more often (F = 6.09, df = 1, *p* = 0.01, η^2^ = 0.01), or 24 h containing their cat (F = 26.1, df = 1, *p* < 0.001, η^2^ = 0.02) compared to cat owners who were currently allowing their cats to 24 h roam (Figure 2).

### 3.2. COM Theme Reliability and Comparisons

All items reflected an adequate internal consistency (Table 1) [39]. Scale scores for each of these themes were computed by averaging the items which were then used for subsequent analysis.

The Capability, Social Opportunity, Cat Welfare Motivation and Community Motivation theme ratings of cat owners who were currently keeping their cat 24 h contained were significantly higher than those of cat owners who were currently practicing a night curfew. Likewise, the ratings of cat owners who were practicing a night curfew were significantly higher than those owners who currently let their cat 24 h roam (Table 3 and Figure 3).

### 3.3. Variables Influencing Cat Free-Roaming Behaviour

A multiple regression was conducted to identify which variables had a significant effect on the time companion cats spent roaming freely (Table 4). Variables added to the model included cat owner age and gender (dichotomous: 0 = not female, 1 = female), the number of cats owned, location (dichotomous: 0 = urban, 1 = regional), type of dwelling (dichotomous: 0 = apartments/units, 1 = standalone house), home ownership (dichotomous: 0 = rent, 1 = own), access to an outside space (dichotomous: 0 = no access, 1 = access available) and the COM themes (Capability, Social Opportunity, Cat Welfare Motivation and Community Motivation; 1 = low agreement, 5 = high agreement).

Three of the COM themes (Capability, Cat Welfare Motivation and Community Motivation) along with the cat owners’ home ownership and type of dwelling predicted significant amounts of the unique variance of time spent by their cats outside, with no restriction on their movements (Table 4). Overall, the final regression model explained 47% of the variance. Cat owners’ capability to contain their cats explained 12% of the unique variance in the regression, while the other two COM items and type of dwelling explained 1% each, and home ownership explained less than 1%.

These results indicate that an increase in cat owners’ capability and motivation (both cat welfare and community framed) to contain their cat will likely reduce the amount of time spent by cats roaming freely, along with living in an apartment or unit. The cats of renters also tended to spend less time roaming freely outside, although these effects were minimal.

### 3.4. Reasons for Allowing Cats to Roam

Cat owners were given the opportunity in an open-ended format to list the factors they have considered when deciding to allow their cat to roam freely (Figure 4). The factor listed by most online respondents was that it was okay for their cat to roam during the day, just not at night. The next most popular factor was that their cat did not roam very far from their property.

### 3.5. Cat Owner Segmentation

To develop the most effective policies and engagement interventions, we not only need to understand why cat owners are willing or not to adopt containment practices, but also if these reasons are similar across all cat owners. Latent profile analysis indicated that cat owners who currently allow their cat to roam away from their property (*n* = 1580) could be classified into six profiles. This solution produced the lowest BIC value, and highest entropy value, with the Lo–Mendell–Rubin test indicating that it fitted the data significantly better than the 7-profile solution (Table 5).

The demographic and behavioural characteristics for each profile are described in Table 6. There were significant differences between the profiles for:All four COM themes (MANOVA F = 326.91, df = 20, *p* < 0.001, η^2^ = 0.49; Capability F = 84.81, df = 5, *p* < 0.001, η^2^ = 0.21; Social opportunity F = 892.98, df = 5, *p* < 0.001, η^2^ = 0.74; Cat Welfare Motivation F = 460.15, df = 5, *p* < 0.001, η^2^ = 0.59; Community motivation F = 242.16, df = 5, *p* < 0.001, η^2^ = 0.86);Age (F = 12.21, df = 5, *p* < 0.001, η^2^ = 0.04);Future intentions (MANOVA F = 70.04, df = 10, *p* < 0.001, η^2^ = 0.18; Prevent from roaming more often F = 99.54, df = 5, *p* < 0.001, η^2^ = 0.24; Keep 24 h contained F = 110.99, df = 5, *p* < 0.001, η^2^ = 0.26);Current containment behaviour (Pearson Χ^2^ = 39.62, df = 5, *p* < 0.001, r = 0.15);Location (Pearson Χ^2^ = 17.43, df = 5, *p* < 0.01, r = 0.10);Gender (Pearson Χ^2^ = 11.04, df = 5, *p* = 0.05, r = 0.07) (Figure 5).

## 4. Discussion

Free-roaming cats are a global challenge, causing nuisance through noise pollution, property damage, urine and faecal soiling and disease transmission [44]. Roaming cats worldwide are also at risk of injury from motor vehicle accidents and animal attacks, as well as being at risk of becoming lost or stolen and contracting infectious diseases. In addition, some regions, including Australia, have wildlife populations that are especially vulnerable to cat predation, making cat containment an important target for human behaviour change interventions. We report a relatively high rate of cat containment compared to previous studies from a large sample of Australian cat owners. This is consistent with the literature over several decades that demonstrates increasing rates of cat containment in Australia, from less than 1 in 4 cats in the 2000s [45], to close to half of all cats in a 2019 study [46], which presumably reflects shifting social norms. Importantly, cat owners who do allow their cats to roam differ and can be divided into six segments, with major implications for the intervention design and delivery. These findings will be applicable to international readers, particularly those from Western countries where companion animal management norms are similar to those in Australia.

A cat owner’s capability to contain their cat had the greatest influence on the amount of time cats spent contained. Cat owners who believed preventing their cat from roaming would be too difficult, were not confident they could prevent their cat from roaming or were not confident they could meet their cat’s needs if not roaming were significantly more likely to allow their cats to roam. Cat owners’ motivation and opportunity also significantly influenced their likelihood to prevent roaming. Cats living in apartments, and cats with younger and female owners were also more likely to be fully contained; however, the owner capability was by far the most important contributor.

Variables reflecting a cat owner’s physical capability to contain their cats, such as location (urban vs. regional), home ownership status and access to outdoor space, did not influence whether cats were prevented from roaming, suggesting that a cat owner’s physical capability is less important to undertaking this behaviour than their psychological capability, i.e., their knowledge or psychological skills, strength or stamina to engage in the necessary mental process [34]. Indeed, we found that cat owners who owned their homes, who lived in free-standing houses (as opposed to apartments), who had access to outdoor space and who had only one cat were significantly more likely to allow their cats to roam; however, these are all scenarios that might be expected to improve a cat owner’s physical capability to contain their cat.

Our findings suggest that audience segments already employing a night curfew for their cats (Concerned Protectors, Conscientious Caretakers and some Laissez-faire Landlords) will be most receptive to full 24 h containment. Two of these segments (Concerned Protectors; Conscientious Caretakers) reported an intention to increase their containment behaviour in the future and hence can be considered the ‘low-hanging fruit’. As quantified by McLeod et al. [33], 24 h containment more effectively reduces the number of free-roaming cats but has a lower likelihood of adoption than a night curfew. Concerned Protectors, while a small group of cat owners, are already highly motivated to keep their cats contained and have the physical capability and opportunity to do so. Hence, this group might be the most receptive to behaviour change techniques focused on education that might be ineffective on their own for other audience segments [27]. These include providing ‘information about social and environmental consequences’, ‘information about health consequences’ and ‘prompts and cues’, for example, to ‘close the door’, ‘bring your cat in’ or ‘check—where is your cat now?’ [34]. Prompts and cues (words or images related to the concept of keeping their cat at home) presented close to the decision point (when feeding their cat, or when they would usually let them in or out) might work well for this group, as they are already committed to the goal of containing their cats and already have the skills and knowledge to perform these behaviours effectively [47].

Conscientious Caretakers were the largest segment and might be the most promising target of behaviour change campaigns. Cat owners in this segment are sensitive to the community impacts of cats roaming (wildlife predation; nuisance to neighbours), are somewhat motivated to contain their cat more in future and are mostly already employing a night curfew. In addition, Laissez-faire Landlords are a group with no firm opinions on cat management and as such might be more open to discussions about management than those with strong, existing viewpoints. However, because of their lack of prior interest they might be relatively difficult to engage initially [21]. This is a group that might be less likely to seek cat management advice or invest resources into cat management. Crowley et al. [21] suggest that this segment might be receptive to prominent, coherent messaging promoting simple, inexpensive and easy-to-implement strategies.

Several audience segments (Tolerant Guardian, Laissez-fare Landlord and Freedom Defender) perceive a 24 h cat containment as having considerable barriers with limited benefits. Those allowing their cats to roam can be strongly motivated by their belief that cats need to roam, and also by a dislike of the smell of cat urine inside [48]. Twenty-four-hour containment is also a behaviour that is difficult to incentivise due to its complexity. Consistent with the findings of this study, previous research has reported that a proportion of Australian cat owners who oppose 24 h containment will agree with a night curfew [26,28,48]. As such, interventions for these audience segments could target a night curfew as a behaviour that is easier to adopt and might act as a catalyst to encourage the more difficult behaviour of 24 h containment in the future [33,48,49].

Our findings suggest that interventions aiming to reduce the roaming of pet cats should target cat owners’ psychological capability to contain their cats. As such, behaviour change techniques focused on education, enablement and training are likely to be most effective [34]. However, interventions that only provide general education content often fail to produce a significant behaviour change [30], and training might be cost-prohibitive to provide at scale, hence, a focus on enablement might be the most practical. Relevant enablement behaviour change techniques include ‘demonstration of the behaviour’, ‘social support’, ‘goal setting’ and ‘action planning’ [34]. A final two enabling behaviour change techniques might be especially valuable in the context of cat containment: ‘adding objects to the environment’—for example, enhancing a cat’s at-home environment with vertical space, scratching surfaces, opportunities for scent marking and predatory play to ensure cat owners can meet all their cat’s behavioural needs at home [50] and a ‘restructuring of the physical environment’ through the construction of secure catios, or modifying fencing.

Interventions for most audience segments (Conscientious Caretaker, Tolerant Guardian, Laissez-fare Landlord and Freedom Defender), whether with 24 h containment or a night curfew as their target behaviours, should aim to encourage cat owners to act while also improving their psychological capability. The behaviour change techniques that might most effectively encourage cat owners in this context, in addition to those discussed above for enablement, include ‘persuasion’, ‘incentivization’ and ‘modelling’, which could be effectively delivered using communications and marketing strategies [34]. Seeing a ‘similar other’ (in this instance, a cat owner like them) modelling the behaviour has been associated with increased self-efficacy and increased engagement in the target behaviour [51]. Incentivisation using competitions might be beneficial to increase engagement without excessive cost, especially as the behaviour change effects from competitions might be the strongest for those initially less motived [52]. Face-to-face exchanges between individuals are likely to be particularly effective [34], especially when using messengers that are most trusted by cat owners such as veterinarians, or RSPCA staff [1,37,46]. Motivational interviewing—a client centred, evidence-based counselling method aiming to strengthen a person’s motivation and commitment to behaviour change [53]—might be applicable in this context but has to-date been underutilised in veterinary practice [54].

Consistent with Crowley et al. [21], our findings suggest that campaigns promoting benefits of containment for cat safety and for wildlife conservation—recommended by previous research [23,28,37]—might align with the values of Concerned Protectors and Conscientious Caretakers, but not with values of other important segments, especially Tolerant Guardians and Freedom Defenders. These segments often have ‘working cats’; they like that their cat hunts and feel strongly about their cat’s need to roam [21]. A different intervention approach is needed for these segments that aligns with their different values and priorities, potentially with messaging focused around protecting wildlife, being a good neighbour and caring for the community.

Interventions focused on increasing the opportunity for cat containment might have the greatest promise for reducing numbers of cats roaming in the medium- and long-term. These interventions aim to decrease the barriers to containment and increase the benefits on a systemic level [34]. Interventions might include making housing more cat-friendly, for example by changing legislation to allow cats in more private rentals and removing regulatory barriers to cat owners modifying fences and constructing cat enclosures. Most importantly, interventions should directly address cat overpopulation. While rates of desexing are high amongst owned pet cats [46,55], close to half of the pet cats in Australia are passively acquired, i.e., acquired for free ‘to give them a home’ [1]. These passively acquired cats, along with cats acquired from animal pounds and shelters, largely originate from the semi-owned and unowned cat population, few of whom are desexed [17]. We did not ask how cats were acquired in this study (whether passively or actively); however, this is an important area for future research. Are those who passively acquire cats more or less likely to allow them to roam? How does the acquisition source relate to target audience segmentation?

McLeod et al. [48] demonstrated that most adopters of cats from an animal shelter in Australia intend to keep their cats contained, which might suggest that owners who actively acquire their cats are more likely to adopt this behaviour. Reducing cat overpopulation, and consequently reducing the proportion of pet cats who are passively acquired, might be important to increase the opportunity for pet cat containment, while also directly reducing numbers of free-roaming cats. Desexing cats has been identified as the most effective intervention to reduce numbers of roaming cats, and is the intervention with the greatest likelihood of adoption [33]. There is likely to be strong support from cat owners and the public for humane strategies for reducing cat populations [21]. In addition, there is a growing body of research on humane and effective cat population management interventions [44].

This online survey sample was not randomly selected and hence likely experienced some sampling bias in favour of people already containing their cats. Consequently, the study might have overestimated the proportion of people fully containing cats in the general population. Future surveys of cat owners might benefit from the use of random online panels to reduce sampling bias.

## 5. Conclusions and Recommendations

It is important to recognise cat owners as key partners in reducing the negative impacts of cats. Cat containment in Australia is increasingly being adopted by cat owners, with more than half the participants in our study already fully containing their cats. Understanding the differences in priorities, values and perspectives of cat owners who are not containing their cats can assist with designing behaviour change interventions. Increasing cat owners’ psychological capability to contain their cats, i.e., their knowledge or psychological skills, strength or stamina, as well as encouraging the adoption of a night-curfew as a first step towards 24 h containment, are recommended. Maintaining a constructive tone and focusing on actions cat owners can take to reduce their cat’s impacts rather than demonizing cat owners is likely to be met with the most success. Concurrent interventions that address cat overpopulation and reduce the number of cats who are passively acquired might be important for increasing cat owners’ opportunity to contain their cats.

## Figures and Tables

**Figure 1 animals-13-01630-f001:**
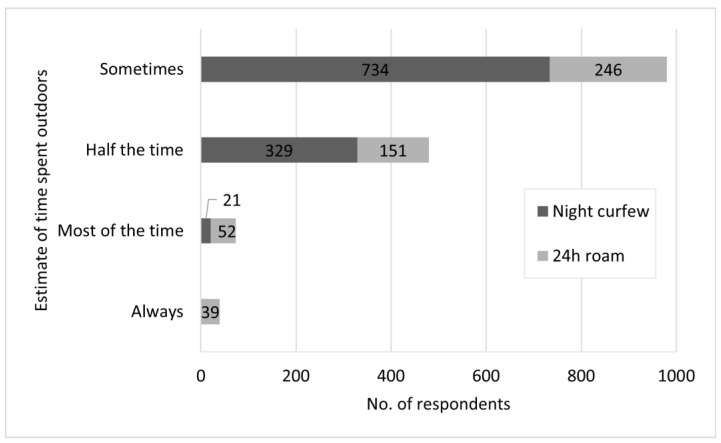
Estimate of time spent outside by cat owners who currently practice a night curfew or let their cats 24 h roam.

**Figure 2 animals-13-01630-f002:**
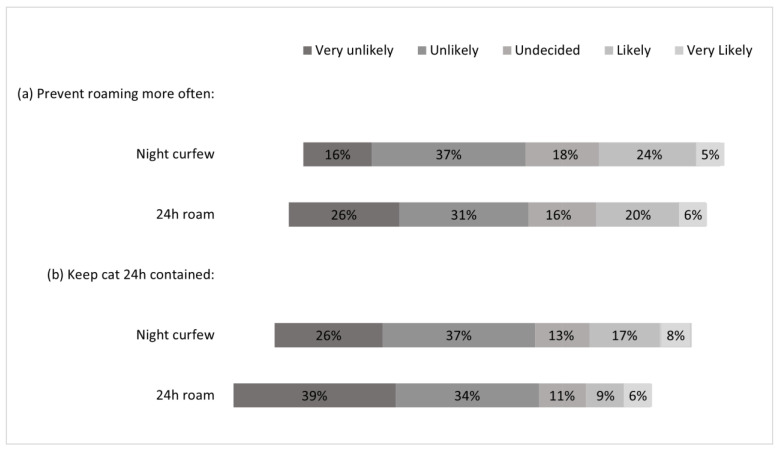
Likelihood of future intentions to (**a**) prevent their cat from roaming more often and (**b**) keep their cat 24 h contained by cat owners who either currently practice a night curfew or let their cats 24 h roam.

**Figure 3 animals-13-01630-f003:**
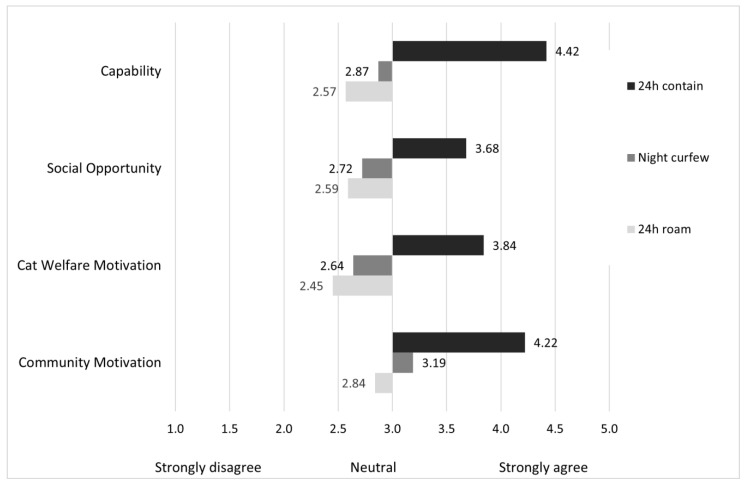
Average agreement of cat owners (*n* = 4482) on ratings to COM themes, of their current cat containment behaviour (24 h contain, night curfew or 24 h roam).

**Figure 4 animals-13-01630-f004:**
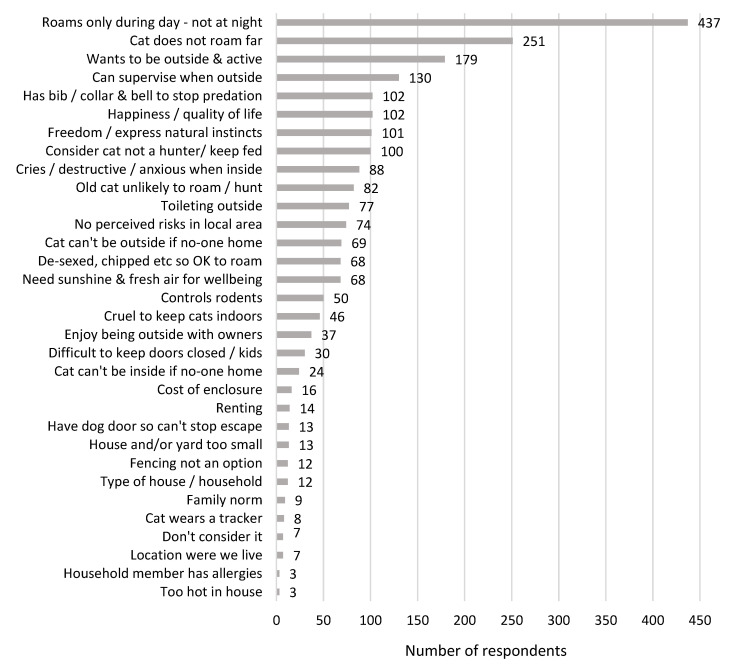
Main factors considered by cat owners when deciding to allow their cat to roam freely.

**Figure 5 animals-13-01630-f005:**
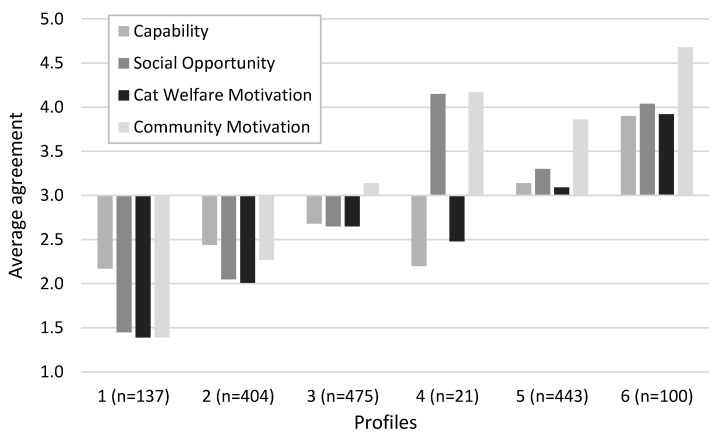
Average agreement scores of the Capability, Social opportunity and Motivation themes across the six identified profiles.

**Table 1 animals-13-01630-t001:** Reliability of COM (Capability, Social Opportunity, and Motivation—cat welfare-framed and Motivation—community-framed) themes and individual items that were rated by respondents.

COM Theme	Individual Items	Cronbach α
		0.82
Capability to contain cat (3 items)	1. Preventing cat roaming is too difficult (reversed score)	
	2. Confident can prevent cat roaming freely	
	3. Confident can provide everything to ensure contained cat is happy	
		0.83
	4. A practice that my family and friends would agree with	
Social Opportunity for cat containment (5 items)	5. A practice that veterinarians would agree with	
	6. A practice that my neighbours would agree with	
	7. A practice that other cat owners would agree with	
	8. Council should have law requiring cats to be kept on owners’ premises	
		0.78
	9. Should be prevented from roaming to keep them safe	
Motivation—cat welfare-framed (3 items)	10. Should be prevented from roaming as good for their health and wellbeing	
	11. Believe cats do not like being contained (reverse score)	
		0.85
Motivation—community-framed (4 items)	12. Should be prevented from roaming to protect wildlife	
	13. Should be prevented from roaming, as can be nuisance to neighbours	
	14. Would prevent from roaming if required by law	
	15. Believe cats should be allowed free to roam (reverse score)	

**Table 2 animals-13-01630-t002:** Comparison of demographic, situational and cat ownership variables across the different containment behaviours of cat owners.

Variables		24 h Contain (*n* = 2896; 65%)	Night Curfew (*n* = 1088; 24%)	24 h Roam (*n* = 492, 11%)	Total
		Mean (SD)	Mean (SD)	Mean (SD)	
Age:		45.2 (±14.0) ^a^	48.9 (±12.4) ^b^	44.8 (±13.5) ^a^	
		*n* (%)	*n* (%)	*n* (%)	
Gender:	Female	2406 (83%)	897 (82%)	377 (77%)	3680
	Male	375 (13%)	136 (13%)	90 (18%)	601
	Non-binary	110 (4%)	50 (5%)	24 (5%)	184
Location:	Urban	2085 (72%)	753 (69%)	335 (68%)	3173
	Regional	811 (28%)	335 (31%)	157 (32%)	1235
Dwelling:	Own	2109 (73%)	891 (83%)	398 (82%)	3398
	Rent	768 (27%)	187 (17%)	89 (18%)	1044
Type of dwelling:	Free-standing house	2000 (69%)	961 (88%)	417 (85%)	3378
	Apartment/other	888 (31%)	127 (12%)	73 (15%)	1088
Outdoor space:	Access	2361 (82%)	956 (88%)	426 (87%)	3743
	No access	527 (18%)	132 (12%)	64 (13%)	723
Cats per household:	1 cat	1451 (51%)	590 (55%)	282 (58%)	2323
	2 cats	967 (34%)	342 (32%)	141 (29%)	1450
	3 cats	251 (9%)	100 (9%)	36 (7%)	387
	4 cats	92 (3%)	25 (2%)	11 (2%)	128
	5 cats or more	112 (4%)	20 (2%)	18 (4%)	150

Notes: Mean for age with different superscripts (in rows) differ significantly at *p* < 0.05 Tukey HSD. Not all respondents provided responses for all questions.

**Table 3 animals-13-01630-t003:** Comparison of agreement with COM themes across the different containment behaviours of cat owners.

	24 h Contain	Night Curfew	24 h Roam			
COM themes	Mean (SD)	Mean (SD)	Mean (SD)	F	*p*	η^2^
Capability	4.42 (±0.71) ^a^	2.87 (±0.92) ^b^	2.57 (±0.93) ^c^	2230.83	<0.001	0.50
Social opportunity	3.68 (±0.74) ^a^	2.72 (±0.81) ^b^	2.59 (±0.80) ^c^	887.93	<0.001	0.28
Cat welfare motivation	3.84 (±0.81) ^a^	2.64 (±0.80) ^b^	2.45 (±0.82) ^c^	1246.06	<0.001	0.36
Community motivation	4.22 (±0.78) ^a^	3.19 (±0.90) ^b^	2.84 (±0.99) ^c^	981.56	<0.001	0.31

Notes: Mean scores for COM themes using scale: 1 = strongly disagree; 5 = strongly agree. Means with different superscripts (in rows) differ significantly at *p* < 0.05 Tukey HSD. η^2^ (partial eta squared) = effect size.

**Table 4 animals-13-01630-t004:** Summary of multiple regression analysis: variables predicting the time spent by cats roaming freely outside.

	95% CL				
Predictors	B	LB	UB	sr^2^	r
Capability	−0.36	−0.38	−0.33	0.12	−0.65 **
Social opportunity	0.01	−0.03	0.04	0.00	−0.48 **
Cat welfare motivation	−0.10	−0.13	−0.07	0.01	−0.55 **
Community motivation	−0.11	−0.14	−0.08	0.01	−0.52 **
Cat owner age	0.00	−0.00	0.00	0.00	0.04 *
Cat owner gender	0.04	−0.01	0.08	0.00	0.05 **
Location	−0.01	−0.05	0.03	0.00	0.01
Type of dwelling	0.16	0.12	0.21	0.01	0.17 **
Access to outdoor space	0.04	−0.01	0.09	0.00	0.06 **
Home ownership	0.06	0.02	0.11	0.00	0.10 **
Number of cats owned	0.01	−0.01	0.02	0.00	−0.03

Model: R = 0.69, R^2^ = 0.47, Adjusted R^2^ = 0.47, F = 337.54 and *p* < 0.001. B unstandardised beta coefficient, CL Confidence limits, LB Lower boundary, UB upper boundary, sr^2^ squared semi-partial correlation (unique variation explained by each predictor), r Pearson correlation, * significant at 0.05 level (2 tailed) and ** significant at 0.01 level (2 tailed). Note R^2^ is not just the sum of all the individual squared semi-partial correlations but also constitutes a portion that is due to within correlation between two independent variables.

**Table 5 animals-13-01630-t005:** Model fit indices for the Latent profile analysis solutions.

Profile Solution	BIC	Entropy	LMR
2	14,439.96	0.81	*p* < 0.001
3	13,905.08	0.79	*p* < 0.001
4	13,702.26	0.78	*p* = 0.03
5	13,610.89	0.77	*p* < 0.001
6	13,595.97	0.81	*p* = 0.04
7	13,606.91	0.76	*p* = 0.33

Notes: BIC—Bayesian information criterion; LMR—Lo–Mendell–Rubin likelihood ratio test.

**Table 6 animals-13-01630-t006:** Demographic and behavioural characteristics of the six cat owner profiles who currently allow their cats to roam away from their property, including how these profiles relate to segments identified by Crowley et al. [21]. NB while Profile 4 differed from Profile 3 in COM agreement scores and demographic variables; we consider both profiles correspond sufficiently with the description of the Laissez-faire Landlord of Crowley et al. [21] to warrant similar approaches to behaviour change interventions.

	Profile 1 (*n* = 137)	Profile 2 (*n* = 404)	Profile 3 (*n* = 475)	Profile 4 (*n* = 21)	Profile 5 (*n* = 443)	Profile 6 (*n* = 100)
	Freedom Defender	Tolerant Guardian	Laissez-Faire Landlord		Conscientious Caretaker	Concerned Protector
Current behaviour	Minimal containment	Minimal containment	Mixture	Mixture	Mostly night curfew	Mostly night curfew
Intentions	No plans to change	No plans to change	No plans to change	Thinking about it more	Thinking about it more	Most likely to change
Average age (years)	Youngest (44.2)	Younger (44.8)	In between (47.1)	In between (49.0)	Older (50.6)	Oldest (51.4)
Location	High urban (75%)	High urban (73%)	Urban (71%)	Urban (67%)	Lowest urban (62%)	Urban (64%)
Gender	Lowest female (72%)	Female (79%)	Female (81%)	Female (76%)	Female (84%)	Highest female (85%)
COM themes	Disagreed most strongly with all COM themes. Members were the least capable of cat containment, did not have the social opportunity, and were not motivated by either cat welfare benefits or community benefits, i.e., believed preventing their cats from roaming would be difficult, perceived roaming as beneficial for cat wellbeing and not a major risk to their safety.	Also disagreed with all COM themes but not to the same degree as Profile 1. Were slightly more capable of containing their cats than Profile 1, did not have the social opportunity, believed cats should be able to roam and were also less concerned about their safety.	No strong opinions about any of the COM themes (they tended not to agree or disagree with any of the drivers).	More likely to be motivated to fully contain their cats for community reasons, i.e., they agreed that cats should not be free to roam and should be prevented from roaming to protect wildlife and prevent nuisance to neighbours. Also demonstrated strong social opportunity to contain their cats, but found containment difficult, with low confidence and skills to contain their cat.	Agreed with community motivation theme, and weak agreement with the remaining COM themes, i.e., motivated by the benefits to the community, both through protecting wildlife and through reducing nuisance for neighbours.	Agreed more strongly with all COM themes, in particular community motivation, i.e., felt capable and had social opportunity to contain cats, were motivated by the benefits to the cats’ welfare and more strongly motivated by the benefits to the community, both through protecting wildlife and through reducing nuisance for neighbours.

## Data Availability

Data supporting reported results can be found at https://www.dropbox.com/scl/fi/lb5vqyg8fnuni4b5wf9gg/OnlineSurvey_CatOwners_NSWRSPCA.xlsx?dl=0&rlkey=ar9ycs2u55of92qyeiuzl505u (accessed on 7 May 2023).

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
