# Peer review of "Understanding the Factors Influencing Cat Containment: Identifying Opportunities for Behaviour Change"

_animals, 2023, doi:10.3390/ani13101630_

Round 1
Reviewer 1 Report
Dear authors,
Congratulations for this study. It is very important and well structured, being a major contribution to the problematic of roaming cats and the wildlife.
1 A brief summary (one short paragraph) outlining the aim of the paper, its main contributions and strengths.
This study is about better understanding the reasons why cat owners decide not to contain their cats from roaming, since it is an important threat to wildlife in Australia and other countries, and how it could be addressed. Although some studies have already investigated cat owners’ perceptions about this issue also using online questionnaires, the current study uses an interesting framework called Capability-Opportunity-Motivation Behavioural model which aligns nicely with the problematic of roaming domestic cats. In addition, the Motivation element was divided in two: cats’ welfare related and community related, representing the two main drivers for owners to contain their cats. Therefore, by classifying owners’ opinions, this framework adds objectivity to the analysis and future interventions which is a major contribution to the field. They found six different profiles of owners whose cats have some degree of free access to outdoors and suggest different strategies for each profile, which is likely to bring about change.
2 General concept comments
2.1 Article: highlighting areas of weakness, the testability of the hypothesis, methodological inaccuracies, missing controls, etc.
The whole article is well organised and clear with sound methodology.
2.2 Review: commenting on the completeness of the review topic covered, the relevance of the review topic, the gap in knowledge identified, the appropriateness of references, etc.
The review is relevant for the topic with appropriate references, showing that there is a lack of knowledge on different styles of cat owners being willing to keep their cats from roaming and thus, how to design interventions for each style in order to be more effective.
The only reference that if included could help increasing the understanding of this issue is Cecchetti M, Crowley SL, Goodwin CE, McDonald RA. Provision of high meat content food and object play reduce predation of wild animals by domestic cats Felis catus. Current biology. 2021 Mar 8;31(5):1107-11, since even though well-fed cats still hunt, the high meat content food might reduce it. However, it is not essential.
3 Specific comments referring to line numbers, tables or figures that point out inaccuracies within the text or sentences that are unclear.
lines 2-3: The title is not very clear, so it takes a little time to understand it. In addition, although this study is very important and applicable for most parts of the world, the discussion focusses on Australia, so Australia could be added in the title. Alternatively, it could be added a paragraph at the discussion comparing the overall results to what is known internationally?
lines 48–51: This sentence is long and thus a little confusing, so it could be beneficial to transform it into two sentences.
lines 51-52: I suggest just mentioning that although "research suggests that most pet cats will hunt if given the opportunity even when well fed”, the composition of the food may play an important role on reducing hunting.
lines 70-71: It would be interesting to reference at least one city/county within Australia that has cat containment or “cat curfews” as mandatory.
line 122: I’m afraid I did not find the inclusion criteria of your participants. E.g., did they have to own a cat at the time they filled out the survey?
line 154: Could you please explain how the dependent variable “hours spent outside freely roaming” was coded? I’m not sure I understood correctly, but looking at the questionnaire I can see that cat owners were asked about frequency of their cats being freely roaming outdoors in specific parts of the day. It is then unclear how this question was translated into an interval variable.
line 158: Is the p value correct in “A significant p value from the LMR test (p=0.05)” or should it be < 0.05?
line 159: Well done for reporting effect size in all your results! Because, unfortunately, it’s not very common in our field, I would suggest mentioning that effect sizes were calculated/reported (η2 and r) in the methodology.
line 164: Were all 4,482 responses complete and used for analysis? If not, how many were incomplete and/or excluded or how did you handle missing data, please?
line 167: I suggest adding a reference or short sentence explaining what “regional areas” in Australia means.
lines 168-170: Are these results in table 2? If so, “table 2” could be mentioned in this paragraph. In addition, the numbers of females and non-binary don’t match.
line 193: Owners without access to outdoor spaces in their properties?
lines 202-203 and Table 1: As mentioned above, it is unclear how this estimation was done based on the questionnaire.
Figure 2: I found its description confusing, so I suggest explaining a little more this figure.
Figure 3: Very informative graph. I just suggest adding more information in its description such as “from cat owners (n= )”.
line 242: “…COM themes…” scores?
lines 248-250: I’m afraid most people do not know that sr2 means how much variation each predictor explains in the regression, so I suggest just mentioning in the sentence “sr2” between parentheses.
Table 4: In the legend, maybe change the order of “r Pearson correlation” with “sr2 squared semi-partial correlation” to keep the same order of the table?
In general, the tables and figures descriptions are quite short. They could explain better what each one is about.
lines 279-289: I’m afraid I haven’t found these analyses described in the methodology. I appreciate they are likely to be the same done in previous analyses, but it would be important to have them stated for comparing the six cat owners’ profiles as well. In addition, have you done ad-hoc analysis to see which profiles differed for each variable? Did you use any type of correction for multiple comparisons? One sentence explaining if you did it or not, and if not, why, would be informative as well.
Table 6: In the current study, is the name “Laissez-faire Landlord” referring to both cat owners’ profiles 3 and 4? If so, it could be explained why it was kept from the previous research and not changed, or if it was decided that five profiles were more suitable. It is confusing the way it is because throughout the discussion the profiles are referred by their “names” which are only five. Maybe adapt the previous nomenclature by adding a variation of the “Laissez-faire Landlord” profile?
lines 385-388: It could be mentioned here that by showing some evidence that the hunting behaviour in cats might be reduced without compromising their welfare, the cat welfare motivation is likely to increase within cat owners (e.g., by changing the cats’ nutrition through a high meat content food and increasing object play, the hunting behaviour was reduced - Cecchetti et al 2021).
Author Response
Thank you for your detailed and thoughtful review. Please find attached our responses to each point.

Reviewer 2 Report
I enjoyed reading this helpful and thoughtful paper. I have only a few concerns.
Throughout the manuscript, presentation of statistical information should be standardized. F values should have df provided (standard APA is F(df) = value). Are the eta squared values partial eta squared values?
110-112: Great idea to make messages relevant to those who will be influenced by them. Reminded me of the work to get all the different types of men who have sex with men to attend to HIV messages.
137: Use colon instead of semi-colon.
148: Should be “Likert scales were”, as “data” is plural.
168: “Ove” should be “Over”;
Table 2: Most of the sample used 24 hr containment.
Figure 1 vs. Figure 2: Why is one left based, and the other free-floating?
247-250: I’m not understanding how the total variance accounted for is 47%, but capability explained 12% and the two motivations and other factors each explained only 1% or less. Something is off here. The authors need to explain this in some way. It doesn’t make sense to the reader.
251-252: It doesn’t make sense to include the two motivations here, as they barely influenced the data.
Table 4: Bottom right, move “-.03” over to the left a bit.
320-322: Put “yet these are” before “all scenarios”. So the authors are only saying these owner factors seem likely to influence cat containment, but according to their data, they don’t; is that correct?
331: On line 79, we were told that “Educating and providing information is rarely enough” to change human behavior. Why would this group be more likely to be influenced by education and information? Explain. Same issue of influence of education on line 361.
348: Change “suggests” to “suggest”, as “et al.” means this is a plural.
355: This idea conflicts with Table 2 in the current study, which indicates most of their participants support 24 hr containment. Are the references referring to random samples? Discuss this apparent contradiction. (The authors indicate the influence of their own convenience sampling on lines 425-327.)
374: “motivation” for much of the paper refers to the data obtained by the authors about the two motivations, but here it seems to mean something different: thus, the authors might want to write, not that motivation should be “increased,” but that things might be done to make the two motivations more influential. I hope that makes sense! This seems to be what the authors are suggesting on lines 395-397, though again the senses of motivation (getting people to do things vs. adhering to the two motivations discussed in the paper) might need to be articulated.
Section 5: Although this section is entitled “Conclusions,” the last two sentences of this final paragraph are not conclusions of the study, but rather hypotheses not dealt with in the study. Perhaps change the name of the section to “Conclusions and suggestions”, or something similar?
Overall, the paper is quite good! My only concern, and it may be due to my lack of statistical knowledge, concerns the apparent contradiction between total variance accounted for and individual variances noted for lines 247-250. I recommend the authors explain this for readers.
Author Response
Thank you for your detailed and thoughtful review. Please find attached our responses to your points.
